# CL-SPO2Net: Contrastive Learning Spatiotemporal Attention Network for Non-Contact Video-Based SpO2 Estimation

**DOI:** 10.3390/bioengineering11020113

**Published:** 2024-01-24

**Authors:** Jiahe Peng, Weihua Su, Haiyong Chen, Jingsheng Sun, Zandong Tian

**Affiliations:** 1School of Artificial Intelligence, Hebei University of Technology, Tianjin 300401, China; 15927009419@163.com (J.P.); haiyong.chen@hebut.edu.cn (H.C.); 13731193988@163.com (J.S.); tianzandong@foxmail.com (Z.T.); 2School of Mechanical Engineering, Hebei University of Technology, Tianjin 300401, China

**Keywords:** contrastive learning, peripheral oxygen saturation (SpO2), computer vision, deep learning, remote photoplethysmography (rPPG), attention mechanism

## Abstract

Video-based peripheral oxygen saturation (SpO2) estimation, utilizing solely RGB cameras, offers a non-contact approach to measuring blood oxygen levels. Previous studies set a stable and unchanging environment as the premise for non-contact blood oxygen estimation. Additionally, they utilized a small amount of labeled data for system training and learning. However, it is challenging to train optimal model parameters with a small dataset. The accuracy of blood oxygen detection is easily affected by ambient light and subject movement. To address these issues, this paper proposes a contrastive learning spatiotemporal attention network (CL-SPO2Net), an innovative semi-supervised network for video-based SpO2 estimation. Spatiotemporal similarities in remote photoplethysmography (rPPG) signals were found in video segments containing facial or hand regions. Subsequently, integrating deep neural networks with machine learning expertise enabled the estimation of SpO2. The method had good feasibility in the case of small-scale labeled datasets, with the mean absolute error between the camera and the reference pulse oximeter of 0.85% in the stable environment, 1.13% with lighting fluctuations, and 1.20% in the facial rotation situation.

## 1. Introduction

With the enhancement of public health, an increasing number of individuals are paying attention to their health status. As intelligent health care advances, remote health services (RHS) are gradually gaining prominence, particularly in regions where medical resources are limited. Moreover, RHS can alleviate patients’ difficulty seeking medical treatment, effectively utilize medical resources, and improve service efficiency. Blood oxygen saturation refers to the ratio of oxygenated hemoglobin in the blood to total hemoglobin. The normal SpO2 of the human body is 95–100%. When the SpO2 value is lower than 80%, it can cause significant impairment to the functions of the heart and brain [1]. Pulse oximeters are primarily used in hospitals, homes, and other places to measure SpO2 values. The sensor of this oximeter emits red and infrared light, a portion of which is absorbed by the blood. The SpO2 levels are measured based on the ratio of photonic currents of red and infrared light at the receiver end. However, this method requires contact with human skin, which may cause discomfort for patients and make it difficult to measure in real time. Another issue with contact pulse oximeters is that obtaining accurate measurements can be challenging for patients with cold hands or circulatory disorders. In nursing homes and among elderly individuals living alone, it is difficult for nursing staff and family members to promptly detect symptoms of fever and other virus-related illnesses in the elderly, which increases the risk of disease worsening. Non-contact monitoring of physiological signs is well suited for caring for premature infants, individuals with disabilities, and emergency personnel. Therefore, remote estimation of SpO2 is a significant area of interest in intelligent medicine. To offer a more comfortable and unobtrusive method for monitoring SpO2, more researchers have investigated SpO2 measurements using videos [2,3]. Estimating blood oxygen levels through videos of the face or other areas of the human skin is based on subtle variations in skin color. Consequently, it is particularly sensitive to environmental noise, which remains difficult to address in this field.

Previous studies have made significant progress in measuring SpO2 physiological signals based on rPPG. This process can be achieved through the ratio method or the deep learning-based method. The ratio method requires using no less than two source-specific wavelengths and multiple cameras. For example, Kong [4] proposed a method for non-contact measurement of SpO2 using light with wavelengths of 520 nm and 660 nm. This method required using two narrow-band filters on the camera to capture the signal and obtain the SpO2 value through algorithm calibration. However, this method faces challenges in wider adoption due to cost considerations. Tamura [5] adjusted the previous ratio method, with light source wavelengths of 660 nm and 940 nm. However, the ratio method ignores the effect of other wavelengths of light, and most studies show that blood oxygen estimation is not a simple linear algorithm. With the development of deep learning and neural networks, convolutional neural networks have played a significant role in feature extraction and signal prediction. Compared with the traditional SpO2 estimation model, deep learning networks use multi-layer architecture to automatically extract deep features from a large set of raw data. Most deep learning-based rPPG methods require large-scale datasets containing facial videos and ground truth physiological signals. Due to the privacy of the medical field, the existing work of SpO2 estimation based on deep learning has the following difficulties: (1) Although facial video is relatively easy to obtain, many videos and physiological signals involving privacy are not disclosed, which makes it challenging to train large-scale data. (2) Some works have proposed deep learning networks to capture multiple features, including spatial, temporal, and periodic features, for SpO2 estimation. However, these models often exhibit limited interpretability and generalization capabilities. (3) Many existing deep learning-based methods primarily use filtering techniques for signal denoising. They do not specifically consider real-world scenarios such as head movements and lighting conditions. This paper designs comparative learning strategies and loss functions tailored to these specific scenarios.

To address the issues above, this paper, inspired by the network presented in [6], introduces a contrastive learning architecture based on 3DCNN. This architecture is designed to extract regions of interest (ROI) from the face and hand in RGB videos, effectively restoring rPPG signals with enhanced interpretability. Additionally, we have developed an attention module grounded in spatiotemporal manual features, which notably improves the efficacy of rPPG feature utilization for blood oxygen estimation. The schematic of our proposed method for video-based SpO2 estimation is illustrated in Figure 1. A detailed exposition of this method is provided in Chapter 3.

The contributions of our work are summarized as follows: (1) This paper aims to resolve the instability and robustness issues due to limited labeled datasets and unstable surrounding environments. An end-to-end innovative semi-supervised method for estimating rPPG multi-channel signals from facial and hand videos is proposed. (2) A lightweight attention block, called a manual feature attention module (MFAM), effectively learns the nonlinear relationships between features. These features are obtained from traditional machine learning-based handcrafted features, ensuring the preservation of the original video components. Additionally, a designed weighting function is employed to integrate the outcomes derived from both learning methodologies. (3) Tests on different datasets show that the proposed method performs well in breath-holding experiments and with good performance under changes in lighting and head rotation movements.

## 2. Related Work

Recent research on non-contact physiological measurements has predominantly centered on remote heart rate (HR) estimation. HR is typically calculated by counting the peak number of rPPG signals over a given period. In comparison, SpO2 estimation represents a more intricate field of study. SpO2 estimation from rPPG signals necessitates precise estimation of signals from different color channels and the extraction of deep features pertinent to blood oxygen levels. Consequently, the video-based blood oxygenation analysis comprises two phases: the initial extraction of physiological signals and the subsequent computation of SpO2, which hinges on the attributes of these physiological signals.

### 2.1. Extraction of Physiological Signals

Time series modeling is a complex endeavor, due to factors such as the volume and quality of data. The complexity is further heightened by modeling temporal dependencies, as well as the critical decision between adopting linear or nonlinear models [7]. The pulse oximeter extracts the pulse wave signal by PPG technology to calculate the SpO2 value. In the process of extracting signals from video taken at a distance, it is necessary to detect the ROI of human bare skin. Previous studies have employed knowledge from the field of machine learning for denoising and purifying remote physiological signals, achieving numerous advancements. Poh et al. [8] pioneered the use of independent component analysis (ICA) [9] to discern pulse waveforms from signals mixed with Gaussian and sub-Gaussian noise. However, the effectiveness of ICA can be constrained in certain scenarios, such as low-light conditions or when the pulsating signal is weak or heavily masked by noise. Lam [10] applied ICA to isolate pulse waveform signals from facial signals, but this method suffered from high computational complexity and prolonged processing time in large-scale data scenarios. In contrast, our study addresses this lighting limitation by optimizing the structure and parameters of the neural network.

Other researchers believed that the subtle color changes produced by different ROIs on the human surface skin are all caused by the heartbeat. Without considering external interference, these signals only differ in amplitude and phase. Therefore, the synchronous components can be separated from the original PPG signals in different ROI regions to achieve the acquisition of pulse waveforms. Tulyakov used an adaptive matrix to select different ROI regions in different video frames for robust rPPG signal extraction in [11]. Vogels proposed a fully automatic living tissue detection method for SpO2 detection during sleep in [12]. Feng analyzed the absorption and reflection of different wavelengths of light such as face motion angle by modeling the face to remove the noise caused by motion artifacts in [13]. An et al. used principal component analysis (PCA) to estimate the palm rPPG signal to solve the palm deception problem in [14]. Sun proposed a method to improve the rPPG waveform using an envelope in [15]. Lu designed a dual adversarial generative network to model the noise in video to extract pure physiological signals from noisy videos in [16]. In contrast, our method posits the physiological signal similarity across different ROIs as a priori, utilizing datasets that account for external interferences to achieve optimal performance.

### 2.2. The Ratio-of-Ratios Principles for SpO2

Early researchers used the Beer-Lambert law [17] to measure the oxygen content in blood without contact. They used the absorption ratio of two different wavelengths of light to estimate SpO2, as shown in Figure 2. There is still a controversy about using the “red and blue channels” or the “red and green channels” as the benchmark. However, in the research of rPPG to estimate SpO2, most researchers use the “red and blue channel”. This is because oxygenated hemoglobin (HbO2) can absorb more blue light than deoxygenated hemoglobin (Hb), and deoxygenated hemoglobin can absorb more red light than oxygenated hemoglobin.

Shao took the lip region as the ROI and estimated the SpO2 values through linear regression of light absorption at various wavelengths in [18]. Scully conducted personalized parameter fitting with different volunteers using a commercial oximeter in [19]. Al-Naji attempted to decompose the Imaging Photoplethysmography (iPPG) signal based on the red and green video channels to extract multi-channel wavelength light features. However, experimental results showed a low correlation between selected red and blue channels and SpO2 concentration in [20]. Lamonaca employed smartphones to measure the fingertips and estimated SpO2 based on changes in light intensity from the red and green channels in [21]. This method requires skin and smartphone flashlight contact, limiting online telemedicine.

Compared with the contact-based SpO2 estimation, Sun [15] proposed a non-contact blood oxygen estimation method using smartphones and introduced a multiple linear regression algorithm. Tarassenko recorded the blood oxygen saturation of dialysis patients in the clinical environment and used the ratio method of signal features of red channel and blue channels to estimate the blood oxygen in [22]. Bal used dual tree composite wavelet transform (DT-CWT) to extract physiological signals from complex signals, to estimate SpO2 saturation robustly in [23]. Guazzi et al. [24] used an RGB camera and an automatic ROI selection algorithm to select the facial region for estimating blood oxygen saturation. Rosa simulated changes in oxygen saturation by performing a breath-holding experiment. They also used Eulerian Video Magnification (EVM) technology to amplify the signal of skin color changes due to the cardiac cycle in [25]. Kim et al. [26] used an RGB camera to capture facial videos under ambient light conditions and converted them into YCgCr color space for SpO2 prediction. However, the researchers have encountered certain limitations in their studies. Most of them have not taken into account the signal drift caused by motion artifacts and variations in light reflection that occur in real-world scenarios. As a result, their findings may have limited applicability in practical applications.

### 2.3. Deep Learning Principles for SpO2

In recent years, deep learning has demonstrated significant efficacy in numerous fields. Previous researchers have achieved improvements in remote physiological indicator measurement through the integration of deep learning methods. KoK et al. [27] utilized the PURE dataset [28] to input the difference features between videos into the adaptive deep network for SpO2 estimation. This method did not account for the impacts over extended periods, performing well in a stable environment. Ding used convolutional neural networks to predict the SpO2 from fingertip videos, demonstrating the potential of 1D convolutional neural networks (1DCNN) in effectively extracting signal features in [29]. Mathew trained three types of neural network models for non-contact estimation of SpO2 in the hand region. They conducted ablation experiments on the model components, which obtained relatively accurate results [30]. The use of hand videos for SpO2 estimation also informed the research methodology of this paper. In this study, we propose a deep learning architecture designed for non-invasive SpO2 monitoring using conventional RGB cameras. This approach exhibits promising prospects for utilization in health screening and telehealth applications.

In this paper, we focused on validating the feasibility of contrastive learning methods for hand and facial regions, thereby enhancing the utilization of individual data and reducing the volume of labeled data required. By analyzing videos simulating real-world scenarios, we aim to adapt the method for daily usage contexts.

## 3. Materials and Methods

### 3.1. Data Pre-Processing

To effectively reduce background noise in videos and meet the requirements of the contrastive learning model, we have employed an ROI approach, focusing on faces and hands for subsequent calculations. For the detection of face and hand regions within the video dataset, we utilized the YOLO object detection algorithm [31]. This recognition algorithm is known for its speed and compact model file size. The process dynamically detected the target position in each frame and scaled the cropped videos to 128×128. The camera was RGBD Camera Intel RealSense D435i, and the video frame rate was 30 frames per second (FPS).

Acknowledging the inherent noise present in the original PPG signals, we have implemented a process to simplify the training challenges. This process involved the application of a band-pass finite impulse response (FIR) filter to the raw PPG signals. The chosen filter had a cutoff frequency range from 0.5 to 4 Hz, effectively encompassing the typical human heart rate spectrum, which ranges from 0.5 to 2.5 Hz. This method preserved essential components like the second harmonic and core signal details in the PPG, eliminating noise and achieving a smoother signal curve. To further enhance the efficiency of our training process, normalization was applied to these filtered PPG signals. Additionally, considering the same frequency as the video dataset, the sampling rate of the PPG signals was adjusted to 30 Hz.

### 3.2. RPPG Contrastive Learning Strategy Based on 3DCNN

The efficacy of 3DCNN in extracting spatiotemporal features from video data is attributed to the concurrent processing of both spatial and temporal dimensions. This enables a more nuanced analysis of dynamic inter-frame changes, thereby facilitating a superior contextual understanding of video content, compared to 2DCNN. This study posited the similar power spectral density (PSD) of rPPG signals in the region of face and hand in a short period as a prior assumption. It is reasonable to assert, based on existing research, that remote estimation of SpO2 and videos from both regions have a strong correlation. From a physiological perspective, there could be a strong correlation between the physiological signals in these two regions. Based on this assumption, we designed a novel training mode to dig into the helpful information. This contrastive learning strategy weakened the influence of facial non-physical features by designing positive and negative samples.

#### 3.2.1. Proposed 3D Convolutional Neural Network

This study designed a 3DCNN architecture to extract features of different channel signals in space and time. After passing through the 3DCNN architecture to achieve spatial information compression and temporal information correlation, a multi-dimensional time series of size 6×T was obtained. The specific 3DCNN structure is shown in Table 1.

#### 3.2.2. RPPG Contrastive Learning Strategy

Previous research indicated that rPPG signals derived from various facial and hand regions exhibit analogous waveforms, with their PSDs demonstrating a comparable pattern. This spatial similarity in rPPG signals has been leveraged in the methodology design of several studies [10,11,32,33,34,35,36]. Despite the minor phase and amplitude variations in rPPG signals from different skin areas of the body [37,38], the transformation of rPPG waveforms into PSD effectively obliterates phase details, and amplitude normalization can be employed to mitigate differences in amplitude. As depicted in Figure 3, the rPPG waveforms from eight spatial regions bear resemblance, converging at the same peak value within their respective PSDs.

In exploring the temporal similarity of rPPG signals, it is acknowledged that HR generally displays a stable pattern over short durations, with minimal rapid fluctuations [39]. This observation was corroborated by Stricker et al. [28], who noted only minor HR fluctuations within brief intervals in their dataset. In an experiment involving the extraction of multiple brief segments from an rPPG clip (e.g., 10 s), the PSDs of these segments showed a remarkable resemblance. For instance, when extracting two segments, each lasting 5 s, from a 10-s rPPG signal and analyzing their PSDs, these demonstrated a notable similarity, particularly characterized by prominent peaks at a consistent frequency.

For the multi-feature extraction process, these video blocks were simultaneously fed into the 3DCNN backbone model, facilitating semi-supervised learning. We set up negative samples during the experiment. During the first T/2 time, the camera simultaneously recorded facial and hand videos, and randomly extracted a frame of image for random noise processing. This process simulated interference in the real environment to enhance the environmental anti-interference performance of the model prediction process. As shown in Figure 4, to establish this internal connection, we designed a unique semi-supervised training mode based on previous related research to obtain signals similar to PPG. Under this premise, in the later T/2 period, slight head rotation was incorporated to enhance the diversity of the dataset, thereby simulating the movements and head rotations that naturally occur in real-world scenarios.

### 3.3. CNN-BiLSTM for SpO2 Estimation

The model of the second stage is the combined structure of CNN and Bi-LSTM, as shown in Figure 5. In tasks like speech recognition and signal processing, the CNN-BiLSTM architecture can effectively identify complex signal patterns. This study represented the application of CNN-BiLSTM in the SpO2 measurement.

### 3.4. Manual Feature Attention Module

Machine learning methodologies have substantiated a robust correlation between the direct current (DC) and alternating current (AC) components of the R, G, and B channels in videos and the SpO2 values estimated by rPPG. In this paper, we designed these manually extracted features as shown in Equation (1) and transformed them via a linear layer and two activation layers to derive Mout2 as shown in Equation (2). As shown in Figure 1, Mout2 obtained by MFAM was subsequently integrated with physiological signals extracted through a 3DCNN for feature fusion.
(1)$$Cw={\left[\begin{array}{cccccc} mean_{red} & mean_{gree} & mean_{blue} & \sigma_{red} & \sigma_{gree} & \sigma_{blue} \end{array}\right]}^{⊤}$$
(2)Mout2=∑i=16(Softmax(ReLU(W×Cw+b)))
where Cw∈ℝ6 denotes the vector of manually extracted features; Mout2 is a scalar, representing the result of blood oxygen feature extraction based on a manual feature attention mechanism; meanred, meangree, and meanblue represent the pixel average values of the three channels in a video of the ROI with a duration of T; and σred, σgree, and σblue represent the variance values of the three channels in a video of the ROI with a duration of T. ReLU is the rectified linear unit activation function. Softmax(⋅) transforms the output of the linear layer into a scalar.

In Equation (3), the fusion function F combines the scalar Mout1, representing the blood oxygen feature output from the deep neural network, with the scalar Mout2, derived from the manual feature attention module, to determine the final SpO2 value.
(3)$$SpO2=F(Mout1,Mout2)=k_{1}Mout1+k_{2}Mout2$$
where $k_{1}$ and $k_{2}$ represent the attention weights assigned to manual features and deep features, respectively, which are autonomously learned by the neural network, satisfying the condition that $k_{1}+k_{2}=100$.

### 3.5. Loss Function

#### 3.5.1. Contrastive Learning Loss Function

As shown in Figure 4, upon architecting our network, the selection of apt loss functions becomes imperative to steer the training process of CL-SPO2Net effectively. To train the contrastive learning part of the method based on the 3DCNN architecture, positive loss $L_{p}$ and negative loss $L_{n}$ were designed. The positive loss function is shown in Equation (4).
(4)$$\begin{array}{cc} L_{p} & =L_{p}^{hand}+L_{p}^{face}+L_{p}^{st}\\ & =\frac{\sum_{n_{1}=1}^{N}\sum_{n_{2}=1}^{N}\sum_{i,j\in U}\left(∥{\left(f_{b}\right)}_{n_{1}}^{i}-{\left(f_{b}\right)}_{n_{2}}^{j}{∥}^{2}\right)+\sum_{n_{1}=1}^{N}\sum_{n_{2}=1}^{N}\sum_{i,j\in V}\left(∥{\left(f_{a}\right)}_{n_{1}}^{i}-{\left(f_{a}\right)}_{n_{2}}^{j}{∥}^{2}\right)}{16N^{2}}+∥f_{a}-f_{b}{∥}^{2}+∥f_{a}-f_{d}{∥}^{2} \end{array}$$where $L_{p}^{face}$ and $L_{p}^{hand}$ were designed based on the time series of facial and hand partitions having similar power spectral densities. $L_{p}^{st}$ represents rPPG spatial similarity and short-term invariance assumptions between the face and hands.

The negative loss function was obtained by comparing the generated pseudo data with real data to eliminate the impact of lighting noise, and the negative loss is defined as
(5)$$L_{n}=-\sum_{n_{1}=1}^{N}\sum_{n_{2}=1}^{N}(∥f_{an_{1}}-f_{cn_{2}}{∥}^{2})/N^{2}$$

#### 3.5.2. Supervised Learning Loss Function for RPPG Signal

Contrastive learning implemented self-supervised learning of rPPG signals. As shown in Figure 4, after training with a large number of unlabeled datasets, small labeled data were then used for fine tuning. The cosine similarity loss function constrains the supervised learning part for 3DCNN. The cosine similarity loss for fine tuning rPPG signals $L_{s}$ is as follows:(6)$$Simiririy(A,B)=\frac{A\cdot B}{\left||A||\times ||B|\right|}=\frac{\sum_{i=1}^{n}\left(A_{i}\times B_{i}\right)}{\sqrt{\sum_{i=1}^{n}A_{i}^{2}}\times \sqrt{\sum_{i=1}^{n}B_{i}^{2}}}$$
(7)$$L_{s}=2-Simiririy(y_{a},{\tilde{y}}_{1})-Simiririy(y_{d},{\tilde{y}}_{2})$$

#### 3.5.3. End-to-End Loss Function for SpO2

Following the results of the first stage of training, we carried out end-to-end training on the entire large model. As depicted in Figure 1, the 3DCNN architecture parameters were frozen and the remaining parameters were trained. The end-to-end loss function is as follows:(8)$$L_{SpO2}=MSE(y_{out},y_{GT})$$

## 4. Results

### 4.1. Datasets

Due to privacy and ethical issues associated with the datasets, a small number of datasets are public. The CL-SPO2Net in this paper utilized datasets comprising FaceForensics++ [40], UBFC [41], and MVPD [42] for both the training and testing phases. The FaceForensics++ dataset encompasses 1000 original videos at 30 fps, which were instrumental for contrastive learning, facilitating the identification of parameter values in proximity to optimal results. The UBFC dataset for supervised learning was compiled using a Logitech C920 HD Pro camera alongside a CMS50E transmissive pulse oximeter, yielding 42 videos coupled with authentic PPG signals. To ensure the efficacy of model training, the multimodal data were uniformly adjusted to a 30 Hz frequency. The MVPD dataset contains videos and physiological information from 13 volunteers, with each volunteer collecting 9 min of video. The high integrity of the MVPD dataset can meet the requirements of neural network models for complete training and validation processes. The dataset is a multimodal database where concurrent videos of the face and hands were utilized for contrastive learning within the methodologies presented in this paper, as well as for fine-tuning mechanisms for rPPG signals. Furthermore, the MVPD dataset supported end-to-end learning for methodologies estimating blood oxygen saturation from videos. The SpO2 collection range in the dataset was 93% to 100%. The examples of the three datasets are shown in Figure 6.

### 4.2. Experimental Setup

The algorithm architecture in this paper was trained using PyTorch, on a Quadro RTX3080 GPU. During the contrastive learning phase, the learning rate was set to 0.001, and the 3DCNN architecture underwent 300 training epochs. In the supervised fine-tuning stage for the rPPG signal, the learning rate was adjusted to 0.0001, with the 3DCNN architecture being trained for 200 epochs. For the end-to-end training phase, the learning rate was configured at 0.005, and the entire model underwent 50 epochs. The true pulse oximetry saturation values were monitored and recorded every second.

### 4.3. Experimental Results

Following the fixation of network parameters for 3DCNN through contrastive learning and signal-supervised training, we employed the MVPD dataset to conduct leave-one-out training and testing on the blood oxygen estimation process. The accuracy of this method was evaluated by metrics Mean Absolute Error (MAE) and Root Mean Square Error (RMSE). The correlation between CL-SPO2Net and video-based blood oxygen estimation results was evaluated by the Pearson correlation coefficient (ρ). The CL-SPO2Net accounted for the effects of motion and lighting variations and contrasted with previous algorithms for estimating blood oxygenation through video, as illustrated in Table 2. The experimental approach included both contact and non-contact video techniques, with ROIs encompassing the fingertip, hand, face, and lip, evaluated under three different experimental conditions. Our method leveraged a larger corpus of unlabeled data for contrastive learning. The experimental results show that the performance of the proposed method approaches the optimum under the influence of factors such as head motion and lighting variations, it exhibits superior performance. In a stable environment, the performance of CL-SPO2Net is close to that of Kim et al. [26].

In breath-holding test experiments, we studied SpO2 measurement using the face region and hand region. Figure 7 showcases the results over a two-minute testing period. Based on the test results of this model, both the accuracy and correlation of video-based oxygen saturation estimation in the facial region are higher than those in the hand region.

We also studied the parameter changes that were manually set during the training process. Table 3 illustrates the respective contributions of the manual feature attention module and the feature extraction of 3DCNN-CNN-BiLSTM to the blood oxygen measurement results. To our knowledge, the model by Rosa et al. [25] employed these manual features for linear regression and validated their relatively reliable correlation. In our study, these manual features resembled MFAM structure, and the features were nonlinearized additionally. Experiments indicate that the effectiveness of deep features surpasses that of manual features.

To further validate the efficacy and rationality of employing contrastive learning for pre-training and the manual feature attention module design in our proposed model, we conducted ablation experiments. These experiments compared various model components against a baseline end-to-end learning method devoid of any novel additions. In the first ablation study, a generic 3DCNN-LSTM module was employed for end-to-end leave-one-participant-out experiments. The second study involved using the 3DCNN and CNN-BiLSTM, as outlined in this paper, for end-to-end leave-one-participant-out experiments. The third study amalgamated the generic 3DCNN-LSTM module with the contrastive learning mechanism proposed in this paper. In the fourth study, the 3DCNN-LSTM architecture was integrated with the manual feature attention module as introduced in this paper. Table 4 presents the medians and IQRs for numerical comparison in the ablation study.

First, we compare the first and the second rows in Table 4. Compared to the baseline, our proposed 3DCNN-CNN-BiLSTM achieves a better MAE with a median of 2.58 and IQR of 0.68, and a better RMSE with a median of 2.95 and IQR of 0.95. In addition, 3DCNN-CNN-BiLSTM achieves a comparable correlation with a better median of 0.51 but a wider IQR of 0.33. The overall better performance of 3DCNN-CNN-BiLSTM suggests the necessity of using the CNN-BiLSTM method. Second, we compare the first and the third rows in Table 4. We observe that Model 1 outperforms the baseline with better medians in terms of correlation (0.62 vs. 0.48), MAE (0.89 vs. 3.57), and RMSE (1.30 vs. 3.98). The overall better performance of Model 1 suggests the necessity of using the contrastive learning strategy. Third, we compare the first and the fourth rows in Table 4. We observe that Model 2 outperforms the baseline with better medians in terms of correlation (0.56 vs. 0.48), MAE (1.42 vs. 3.57), and RMSE (1.86 vs. 3.98), but a wider correlation IQR of 0.36. The overall better performance of Model 2 suggests the necessity of using the manual feature attention module. Last, we observe that CL-SPO2Net outperforms other models with better medians in terms of correlation, MAE and RMSE, and narrower IQR of correlation. This suggests that the module combination we designed can effectively improve accuracy.

In the experiments conducted on the test dataset, we observed that the noise was introduced into the original signals due to facial rotations and fluctuations in lighting. Optimal parameters of 3DCNN were employed to extract the six rPPG signals from the video data. In contrast, our method effectively mitigated this noise by contrastive learning strategy (see Figure 8).

## 5. Discussion

Targeting unobtrusive physiological measurements, unmanned rescue, remote surgery, and other health care domains, this study provides an efficient method for utilizing unlabeled data. This paper offers new perspectives for research in the medical field and other ethically sensitive areas. Compared to the non-contact methods for measuring heart rate, the techniques for estimating SpO2 are considerably more susceptible to interference from noise. CL-SPO2Net effectively utilized unlabeled FaceForensics++ data for learning and employed multimodal information from MVPD and data augmentation techniques. The method is based on physiological information-driven contrastive learning, excluding signals irrelevant to physiological information. The accuracy of physiological estimation, including SpO2 estimation, often depends on whether the noise signal can be effectively removed. The ratio method used a combination of the 6 features in this paper for linear fitting and may not be able to simulate complex SpO2 estimation models. In contrast, MFAM employed manual features as inputs to an attention module in deep learning. This study measured SpO2 with an RGB camera. The results of Kim et al. [26] with YCgCr color space for feature conversion are better than ours only in a stable environment. On a server equipped with a Quadro RTX3080 GPU and based on the YOLO algorithm for facial recognition, CL-SPO2Net performed SpO2 estimation at 0.5 Hz on the computer server with a Quadro RTX3080 GPU. By reducing the resolution and duration of the video, this program was deployed on a home laptop with a 4G GPU and came close to reference measurements.

There are several factors that can be improved in the future. We extracted 6 manual features, all of which are RGB channels. These may have limitations, and the YCgCr color space will be considered in the future. The training and testing processes are all conducted on the server. We only made a demo on a laptop. It is necessary to lightweight the model to reduce the hardware load and adapt to the needs of a wider range of scenarios. Additionally, this paper involved SpO2 levels of 93–100%. It will be necessary to further reduce SpO2 levels and develop a proposed method to expand the measurement range.

## 6. Conclusions

This paper introduces CL-SPO2Net, a semi-supervised neural network. This network consists of a fusion of two methods. Observing the weights of the fusion functions in the two methods, we find that deep contrastive learning contributes more significantly to the estimated results. We have designed breath-holding tests on different ROIs in videos to validate the hypothesis and have found that the face videos have higher accuracy and sensitivity in blood oxygen estimation than hand videos. We have also shown the explainability of our network by visualizing the deep rPPG features in the network. Despite external disturbances, the rPPG signals remain stable, enabling SpO2 estimation to adapt to complex environments. Comparing the measured SpO2 with the reference SpO2 data, the MAE was 0.85% in the stable environment, 1.13% with lighting fluctuations, and 1.20% in the facial rotation situation. The RMSE was 1.12% in the stable environment, 1.37% with lighting fluctuations, and 1.25% in the facial rotation situation. The Pearson correlation coefficient was 0.76 in the stable environment, 0.79 with lighting fluctuations, and 0.80 in the facial rotation situation. These results suggest that the SpO2 levels measured by the introduced video-based method align with those obtained through traditional sensor methods. This confirms the potential for precise SpO2 measurements without the need for direct sensor attachment. In future work, we aim to explore ways to further reduce motion artifacts and noise caused by illumination changes in non-contact physiological measurements. In addition, subsequent research efforts will be directed toward enhancing the speed at which rPPG signals are extracted for the measurement of oxygen saturation levels.

## Figures and Tables

**Figure 1 bioengineering-11-00113-f001:**
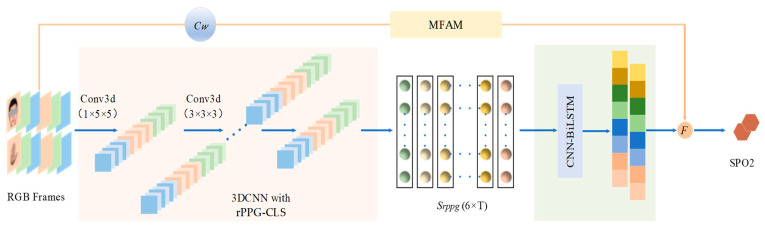
Proposed SpO2 analysis architecture using CL-SPO2Net. Small blocks with various colors represent the features extracted at running steps of the network. The light pink area represents the 3DCNN, within which we combine unsupervised contrastive learning with supervised label learning to obtain accurate rPPG signals for remote videos. The light green area represents the spatio-temporal feature extraction step of the signal. The results obtained by the CNN-BiLSTM network, alongside those from an attention module, are inputted into a fusion function to yield the estimation results of blood oxygen saturation.

**Figure 2 bioengineering-11-00113-f002:**
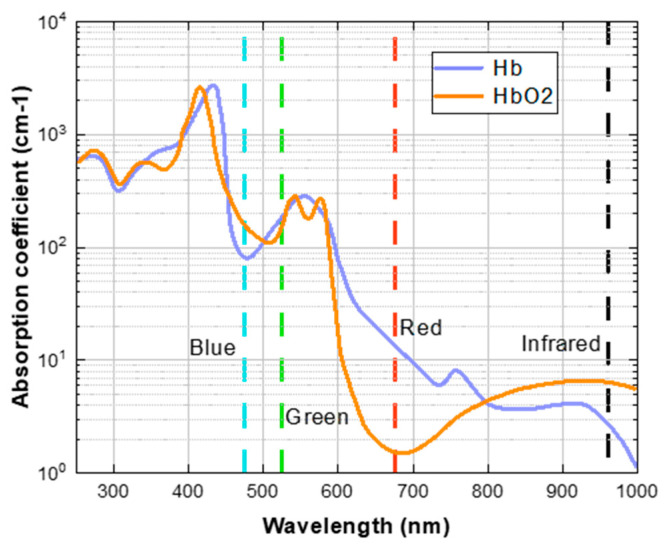
The absorption degree of Hb and HbO2 to light of different wavelengths. The blue, green, and red dashed lines denote the wavelengths of light corresponding to their respective colors.

**Figure 3 bioengineering-11-00113-f003:**
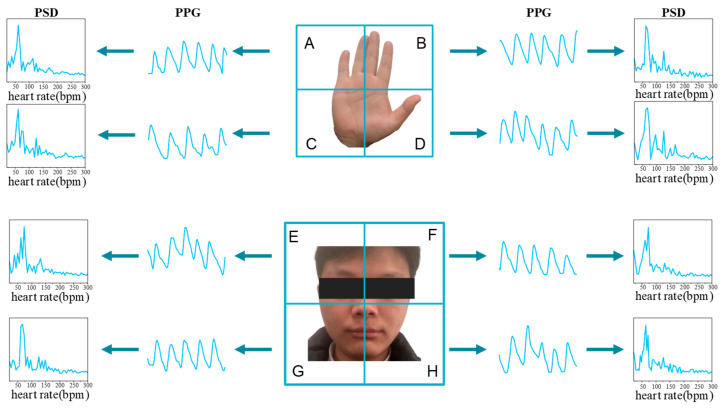
The hypothesis of Similar PSD of facial and hand rPPG signals. Each part is divided into four ROIs, in which the human hand is divided into four regions A–D according to the center line, and the facial part is divided into four regions E–H according to the center line. Set $U=\left\{A,B,C,D\right\}$, $V=\left\{E,F,G,H\right\}$.

**Figure 4 bioengineering-11-00113-f004:**
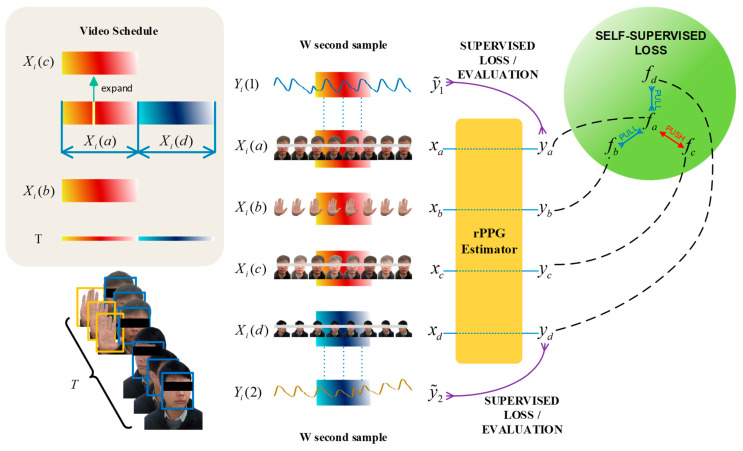
Combination of contrastive learning and supervised learning. Firstly, during the experimental procedure, video segments designated as $X_{i}(a)$ and $X_{i}(b)$ were acquired, capturing the facial and hand regions of a participant who maintained a stationary posture throughout the recording phase. The video clip $X_{i}(c)$ constituted a regenerated sequence derived from video $X_{i}(a)$, wherein a single frame was extracted and subjected to chromatic data augmentation. The video segment $X_{i}(d)$ represented a subject executing a facial rotation maneuver. Time series $x_{a}$, $x_{b}$, $x_{c}$, and $x_{d}$ are the tensor form of video clips of time length T/2. Signals $y_{a}$, $y_{b}$, $y_{c}$, and $y_{d}$ are rPPG signals produced by an rPPG estimator utilizing a 3DCNN architecture, subsequently transformed via Fourier analysis to generate PSD vectors $f_{a}$, $f_{b}$, $f_{c}$, and $f_{d}$, where ${\tilde{y}}_{1}$ and ${\tilde{y}}_{2}$ are the real PPG signals collected from the same object at the time of first T/2 and last T/2.

**Figure 5 bioengineering-11-00113-f005:**
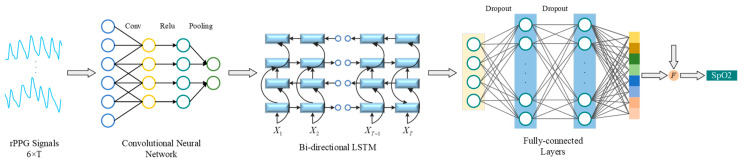
The CNN-BiLSTM architecture.

**Figure 6 bioengineering-11-00113-f006:**
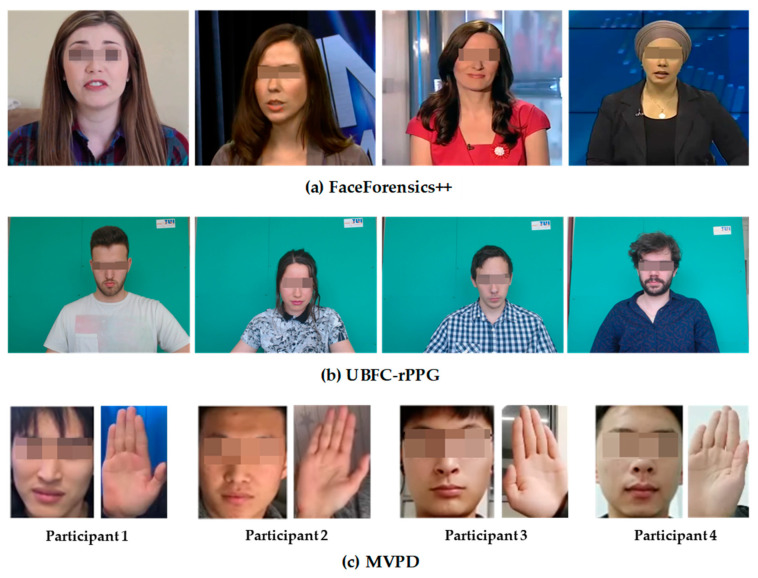
Example images from the datasets used in this paper: FaceForensics++ (**a**), UBFC-rPPG (**b**) and MVPD (**c**).

**Figure 7 bioengineering-11-00113-f007:**
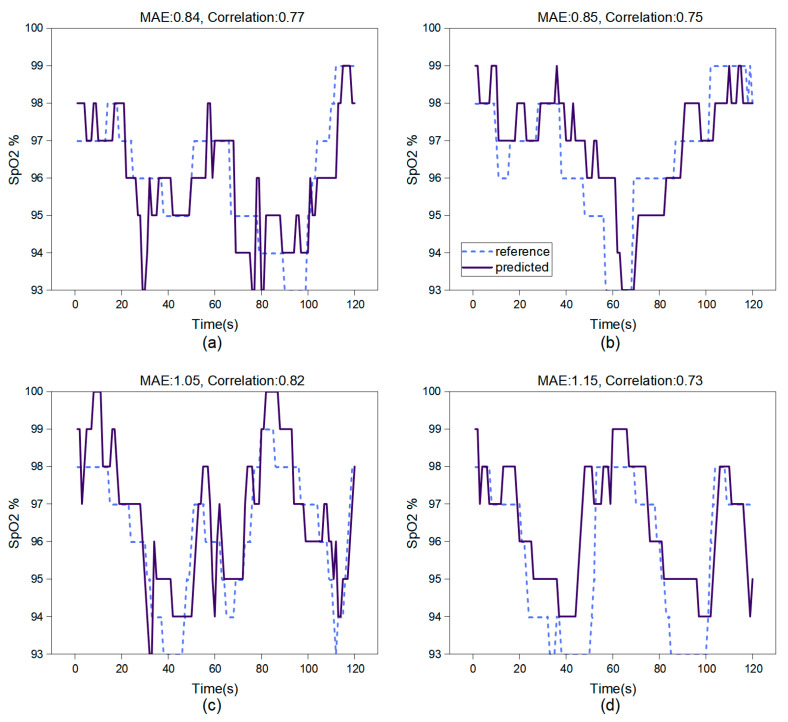
The test results of the CL-SPO2Net on 2 participants. (**a**,**b**) show the face test results; (**c**,**d**) show the hand test results. This network architecture performs SpO2 estimation at 0.5 Hz on the computer server with a Quadro RTX3080 GPU.

**Figure 8 bioengineering-11-00113-f008:**
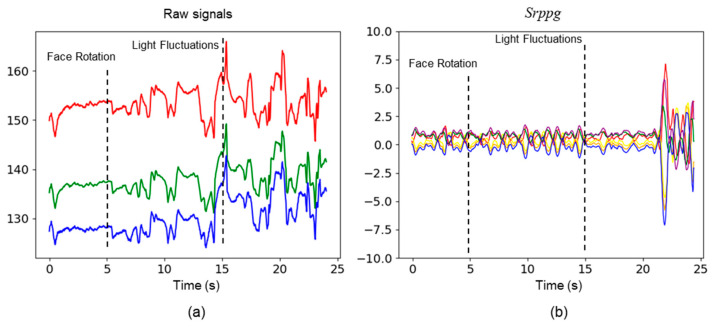
Comparison of raw physiological signals and signals generated by rPPG contrast learning strategy based on 3DCNN. (**a**) shows the raw signals of the red, green, and blue channels; (**b**) shows the time series of six channels, each represented by different colors.

**Table 1 bioengineering-11-00113-t001:** The 3DCNN Structure. Convolutional blocks are shown in brackets, with the number of blocks stacked. We denote filter shape using (time length × height × width, temporal dilation rate) for convolutional layers, (time length × height × width) for pooling layers, and output shape using (output channels × time length × height × width). The time series interpolation used a bilinear interpolation algorithm.

Layers	Blocks	Output Size
Input	standardization	3×T×128×128
3DConv1	1×5×5 stride (1,1,1)	32×T×128×128
Avgpooling1	1×2×2, stride (1,2,2)	32×T×64×64
3DConv2	$\left[\begin{array}{l} 3\times 3\times 3\\ 3\times 3\times 3 \end{array}\right]\times 2$	64×T×64×64
Avgpooling2	2×2×2, stride (2,2,2)	32×T/2×32×32
3DConv3	$\left[\begin{array}{l} 3\times 3\times 3\\ 3\times 3\times 3 \end{array}\right]\times 2$	64×T/2×32×32
Avgpooling3	2×2×2, stride (1,2,2)	64×T/2×16×16
3DConv4	$\left[\begin{array}{l} 3\times 3\times 3\\ 3\times 3\times 3 \end{array}\right]\times 2$	64×T/2×16×16
Interpolate1	-	64×T×16×16
3DConv5	$\left[\begin{array}{l} 3\times 3\times 3\\ 3\times 3\times 3 \end{array}\right]\times 1$	64×T×16×16
Avgpooling4	1×16×16	64×T×1×1
3DConv6	$\left[\begin{array}{l} 3\times 3\times 3\\ 3\times 3\times 3 \end{array}\right]\times 1$	6×T×1×1

**Table 2 bioengineering-11-00113-t002:** The result of related works on the MVPD dataset.

Method	Non- Contact?	ROI	Stable Environment			Light Fluctuations			Face Rotation		
			MAE (%)	RMSE (%)	ρ	MAE (%)	RMSE (%)	ρ	MAE (%)	RMSE (%)	ρ
Ding et al. [29]	✗	Fingertip	2.24	2.56	0.52	-	-	-	-	-	-
Nemcova et al. [43]	✗	Fingertip	1.10	1.23	0.43	-	-	-	-	-	-
Kong et al. [4]	✓	Face	1.39	1.75	0.71	2.13	2.27	0.45	1.83	1.71	0.69
Scully et al. [19]	✓	Face	1.28	1.84	0.85	1.86	2.11	0.74	2.30	2.51	0.57
Shao et al. [18]	✓	Around the Lip	1.05	1.32	0.93	2.35	2.62	0.61	2.55	2.89	0.58
Kim et al. [26]	✓	Face	0.54	0.69	0.92	1.99	1.93	0.71	1.58	1.98	0.79
Sun et al. [15]	✓	Hand	1.13	1.23	0.91	1.55	2.03	0.78	-	-	-
Mathew et al. [30]	✓	Hand	1.97	2.32	0.40	-	-	-	-	-	-
CL-SPO2Net	✓	Face	0.85	1.12	0.76	1.13	1.37	0.79	1.20	1.25	0.80

✓ represents a non-contact scenario, while ✗ denotes a contact scenario.

**Table 3 bioengineering-11-00113-t003:** The evolution of key parameters during the CL-SPO2Net training.

Epoch	$k_{1}$	$k_{2}$	Validation MAE (%)	Validation RMSE (%)
0	50	50	83.05	91.72
10	56	44	42.33	46.97
20	63	37	22.01	25.98
30	71	29	8.36	10.30
40	80	20	1.65	1.96
50	87	13	1.26	1.52

**Table 4 bioengineering-11-00113-t004:** Results of the ablation studies. The performance in the leave-one-out experiments is assessed using Median and Interquartile Range (IQR) as analytical measures.

Method		ρ	MAE (%)	RMSE (%)
Baseline (3DCNN-LSTM)	Median	0.48	3.57	3.98
	IQR	0.32	1.21	1.38
Proposed 3DCNN and CNN-BiLSTM	Median	0.51	2.58	2.95
	IQR	0.33	0.68	0.95
Model 1: Baseline + Contrastive Learning	Median	0.62	0.89	1.30
	IQR	0.31	0.52	0.63
Model 2: Baseline + MFAM	Median	0.56	1.42	1.86
	IQR	0.36	0.77	1.02
CL-SPO2Net	Median	0.76	0.88	1.23
	IQR	0.29	0.54	0.72

## Data Availability

Data associated with this article can be found in the online version, at https://github.com/puterh/rppg-spo2 (accessed on 20 November 2023.).

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
