# Peer review of "CL-SPO2Net: Contrastive Learning Spatiotemporal Attention Network for Non-Contact Video-Based SpO2 Estimation"

_bioengineering, 2024, doi:10.3390/bioengineering11020113_

Round 1

Reviewer 1 Report

Comments and Suggestions for Authors

The problem statement need to be stated clearly in introduction

please show the spo2 rate along side the example image data

please add a section about your research objective

using what basis that you can say your proposed method achieved a good result? can it be implemented in real world?

please separate the discussion and conclusion into its own section

please add your result summary in conclusion section

Comments on the Quality of English Language

The quality is good although some part need to be checked such as "To address the above issues, inspired by the network [6]" use conjunction "and"

Author Response

  1. We gratefully appreciate for your valuable suggestion. The introduction section is crucial for readers to understand the field of the article and enhance their interest in reading. It needs to introduce the necessity and difficulties of research in this field and propose the innovation of the article. We have improved and logically reconstructed the problem description part of the introduction. The first paragraph in the introduction explains the background, necessity and common problems encountered in this research topic. Blood oxygen estimation in the form of non-contact video is a current scientific research hotspot and trend. The second paragraph describes the limitations and device dependencies of video-based blood oxygenation regression using linear regression methods from machine learning. Paragraph 3 presents the problems currently encountered in using deep learning methods to study this topic. By analyzing the importance and difficulties of scientific research in this field, we propose an innovative comparative learning network model architecture. The last paragraph of the introduction explains the main innovative work of the paper, which has an important role in this field.
  2. The sentence “This network architecture performs SpO2 estimation at 0.5Hz on the computer server with Quadro RTX3080 GPU.” has been added in the footnotes of Figure 7. Regarding the SpO2 estimation rate using video, we elaborate in the Discussion Chapter. The training and testing processes are all conducted on the server. We only made a demo on a laptop. It is necessary to lightweight the model for reducing the hardware load and adapting to the needs of a wider range of scenarios.

  3. We have included the research objectives in the related work section. Thank you so much for your careful check. We feel sorry for our carelessness. 
  4. Thank you for your insightful queries regarding the efficacy and real-world applicability of our proposed method. I am pleased to address these points. We have employed several statistical and performance metrics, such as Mean Absolute Error (MAE), Root Mean Square Error (RMSE) and Pearson correlation coefficient, widely recognized in our field, to evaluate the effectiveness of our method. We have compared our method against other leading techniques in our field. This comparison demonstrates the superiority of our approach in key performance metrics. In addition, we added key parameter experiments in Table 3 and feature visualization experiments in Figure 8, and improved ablation experiments in Table 4.

    Feasibility of Real-World Implementation:

    Experimental Design Considerations: Our experimental setup was designed with real-world complexities in mind. For instance, we tested the robustness and stability of our method under various environmental conditions.

    Technical Feasibility Analysis: In the paper, we discuss the practicality of implementing our technique, including the required hardware and software resources, the cost of implementation, and the ease of operation. In addition, regarding the issue of hardware cost, we elaborated on the application prospects of model lightweighting in the discussion and conclusion sections.

    Application Prospects: We also explore the potential applications of our method in specific sectors, such as healthcare and remote monitoring, providing examples of practical use cases.

  5. Thank you so much for your careful check. Please check my revised manuscript.

  6. Thank you so much for your careful check. We have added our result summary in conclusion section. We feel sorry for our carelessness. Please check our revised manuscript.

Reviewer 2 Report

Comments and Suggestions for Authors

The article is based on Contrastive Learning Spatiotemporal Attention 2 Network for Non-contact Video-based SpO2 Estimation. The manuscript is well written and presented,I have minor obsevations on method and results.

Methodology section need to revise by elaborating more on the proposed method. share the simulatiuon parameters

Results need to be added more with more discussion and use cases,

a comparative analysis is important to embedded

Some recent works can be added to enhance the literature section like "A Simulation Model to Reduce the Fuel Consumption through Efficient Road Traffic Modelling"

The abstract should be more concise, few typos need to fix in the manuscript

overall, a good work in the domain

Comments on the Quality of English Language

Okay

Author Response

  1. We gratefully appreciate for your valuable suggestion. We adjusted some details of the methodology section. Figure 1 in the introduction section gives an overview of our method. In the methodology section, we elaborate on the data processing, module decomposition, strategy implementation details, loss function and other parts for detailed explanation. During the elaboration process, we link the full text. The personalization variables and formulas in our approach are explained. Since this method is a neural network structure, in order to further understand the effectiveness and credibility of the method, we added evolution experiments of key parameters, as shown in Table 3. And the experimental setup section explains the details of the software and hardware parameters.
  2. Thank you so much for your careful check. We added key parameter experiments in Table 3 and feature visualization experiments in Figure 8, and improved ablation experiments in Table 4. The explanation and comparative analysis of results were revised in this results section. We revised the discussion and conclusion section for further analysis. Please check our revised manuscript.
  3. I have read the paper "A Simulation Model to Reduce the Fuel Consumption through Efficient Road Traffic Modelling" carefully. This is a very good work. And we think it is very necessary to cite this work. We have cited this work in the related work section.

  4. Thank you so much for your careful check. The abstract requres contextual understanding, clear explanation and concise presentation. We have revised the content of the abstract. Few typos have fixed in the manuscript. Please check our revised manuscript.

Reviewer 3 Report

Comments and Suggestions for Authors

Thanks for giving me an opportunity to review this paper. This study has proposed a contrastive learning spatiotemporal attention network (CL-SPO2Net), an innovative semi-supervised network for video-based SpO2 estimation. This is a good study though but needs extensive editing. Moreover, they have provided lots of details, all are not required in the paper. It is a scientific paper, not a student thesis. Please revise it. 

Comments on the Quality of English Language

Need extensive editing. 

Author Response

We feel sorry for the inconvenience brought to the reviewer. Also thank you so much for your careful check. I have revised the article extensively, including the abstract, introduction, methodology, results, discussion, and conclusion. We have improved the irregularities in English expressions, deleted some unnecessary explanations and explanations in the methodology section, and enhanced the academic quality. Please check our revised manuscript. Thank you again for your valuable comments.

Round 2

Reviewer 3 Report

Comments and Suggestions for Authors

Thanks for new version.

Comments on the Quality of English Language

Extensive English editing is needed.

Author Response

Thank you very much for the comment. The English language in the revised manuscript has been carefully corrected to improve grammar and readability. Errors in detail about units and the problem of missing spaces have been corrected. The paper has already been polished by an established expert in English. Please check if the amended version meets the standard of English presentation.